# The Effect of Processing Route on Properties of HfNbTaTiZr High Entropy Alloy

**DOI:** 10.3390/ma12234022

**Published:** 2019-12-03

**Authors:** Jaroslav Málek, Jiří Zýka, František Lukáč, Monika Vilémová, Tomáš Vlasák, Jakub Čížek, Oksana Melikhova, Adéla Macháčková, Hyoung-Seop Kim

**Affiliations:** 1UJP PRAHA a.s., Nad Kamínkou 1345, 156 10 Prague-Zbraslav, Czech Republic; zyka@ujp.cz; 2CTU in Prague-Faculty of Mechanical Engineering, Karlovo Náměstí 13, 121 35 Praha 2, Czech Republic; 3Institute of Plasma Physics CAS, 182 00 Praha 8, Czech Republic; lukac@ipp.cas.cz (F.L.); vilemova@ipp.cas.cz (M.V.); 4Faculty of Mathematics and Physics, Charles University, 180 00 Praha 8, Czech Republic; tomas.vlasak@seznam.cz (T.V.); jakub.cizek@mff.cuni.cz (J.Č.); Oksana.Melikhova@mff.cuni.cz (O.M.); 5Faculty of Materials Science and Technology, VŠB-Technical University of Ostrava, 17. Listopadu 15, 708 33 Ostrava 8, Cech Republic; adela.machackova@vsb.cz; 6Department of Materials Science and Engineering, POSTECH, Pohang 790-784, Korea; hyoungseopkim@gmail.com

**Keywords:** high-entropy alloy, powder metallurgy, plastic deformation, microstructure

## Abstract

High entropy alloys (HEA) have been one of the most attractive groups of materials for researchers in the last several years. Since HEAs are potential candidates for many (e.g., refractory, cryogenic, medical) applications, their properties are studied intensively. The most frequent method of HEA synthesis is arc or induction melting. Powder metallurgy is a perspective technique of alloy synthesis and therefore in this work the possibilities of synthesis of HfNbTaTiZr HEA from powders were studied. Blended elemental powders were sintered, hot isostatically pressed, and subsequently swaged using a special technique of swaging where the sample is enveloped by a titanium alloy. This method does not result in a full density alloy due to cracking during swaging. Spark plasma sintering (SPS) of mechanically alloyed powders resulted in a fully dense but brittle specimen. The most promising result was obtained by SPS treatment of gas atomized powder with low oxygen content. The microstructure of HfNbTaTiZr specimen prepared this way can be refined by high pressure torsion deformation resulting in a high hardness of 410 HV10 and very fine microstructure with grain size well below 500 nm.

## 1. Introduction

High entropy alloys (HEAs), complex concentrated alloys (CCAs), and multi-principal element alloys (MPEAs) are the most common names of a new group of materials [1,2,3,4,5] introduced by Yeh et al. [6] and Cantor et al. [7] in the beginning of this century. Many definitions were proposed for such materials, but in general they consist of multiple (usually at least five) elements in equiatomic or near-equiatomic composition [2]. This approach is different from the traditional alloy design using one (or maximum two) principal element and other minor elements. The presence of many elements in equiatomic (or near) composition leads to high mixing entropy and thus to many interesting properties [2,5]. One of such effects is the existence of a random solid solution and simple body centered cubic (bcc) or faced centered cubic (fcc) structure. Recently HEAs with hexagonal closed packed (hcp) structure have been reported as well [8,9]. A new approach to HEA design using the valence electron concentration (VEC) was recently proposed with the aim of increasing the plasticity of these alloys [10,11]. Other authors showed that decreasing the solid solution phase stability promotes the stress-induced transformation during loading and therefore may result in increased plasticity of the alloy [12,13,14].

The group of HEAs has the potential to overcome currently used materials in many applications. Some of them have excellent high temperature properties and are therefore candidates for replacement of Ni-superalloys, which are currently used for turbine blades, etc. [15]. Cryogenic applications are also potential fields of interest as some HEAs retain high plasticity even at cryogenic temperatures [5].

In this work the HfNbTaTiZr (equiatomic) alloy was studied. This alloy has the potential to be used in high temperature applications or as a biocompatible material due to its chemical composition consisting of refractory metals [16,17,18], which are biocompatible [19,20,21] elements. This alloy is usually prepared by an arc melting process [22,23,24,25,26]. Power metallurgy was studied as an alternative route of HEA synthesis [27,28,29,30]. Mechanical alloying (MA) followed by hot isostatic pressing (HIP) [30] or spark plasma sintering (SPS) [27,28] are the most common powder metallurgy techniques. HfNbTaTiZr alloy was recently produced from commercially available elemental powders that were cold isostatically pressed (CIP) and sintered. Commercially available elemental powders were used for TiNbTa_0.5_Zr and TiNbTa_0.5_Zr_0.2_Al alloy synthesis by Cao et al. [29]. This way of preparation may overcome the main drawbacks of the arc melted alloys, which suffer from very coarse dendritic microstructures. The dendritic microstructure may lead to phase separation (a mixture of two phases with slightly different chemical composition) due to variation of chemical composition in dendrite and interdendritic regions [23,31]. The microstructure of specimens produced by powder metallurgy exhibits no dendrites and also the grain size is on average significantly finer than in the arc melted samples. On the other hand, residual porosity present after sintering deteriorates the mechanical properties of HEAs significantly [29,32,33]. This is probably the main drawback of powder metallurgy processed HEAs. Moreover the initial powders have large active surfaces and could therefore be vulnerable to oxidation. Hot forging deformation can be one of the ways to eliminate the porosity. Hence, forging may in general improve microstructural and mechanical properties of HEAs [34]. Our previous experiments showed that the residual porosity of as-sintered HfNbTaTiZr specimens was eliminated during hot compression tests. However, it should be pointed out that the hot compression tests and real forging processes may significantly differ from each other. Therefore in this work various techniques of removing residual porosity were examined. Microstructure and mechanical properties of HfNbTaTiZr alloy prepared by various methods of powder metallurgy are reported in this paper. The possibility of using such techniques for bigger specimens under real conditions was accentuated as well. 

## 2. Materials and Methods

Several methods of powder metallurgy were used to prepare the equimolar HfNbTaTiZr HEA. Preparation routes of various samples are summarized in Table 1. Elemental powders (purity >99%) and granularity of −325 mesh (<44 μm) were supplied by Huarui Co. (Chengdu, China). Weighting and mixing of powders was performed under Ar protective atmosphere. Three different kinds of initial powders were used: (i)elemental powders mixed in appropriate ratio in Turbula 2F device for 10 h at 45 rpm and subsequently pressed into green compacts (denoted here HEAP) using CIP under the pressure of 400 MPa;(ii)MA powder prepared from the elemental powders (granularity −325 mesh) by high energy ball milling in an Ar atmosphere for 42 h using tungsten carbide balls. The mean particle size of the MA powder was ≈3 μm and MA particles consisted of nanocrystalline grains (iii)atomized powder (AT) prepared from arc melted HfNbTaTiZr alloy by crucible-free electrode induction-melting gas atomization in a protective Ar atmosphere to suppress undesirable oxidation. The AT powder had a broad particle size distribution covering the range from 10 to 300 μm.

Green compacts (HEAP) were further processed via different routes. Sintering at 1400 °C for 16 h was carried out in a vacuum furnace and sintered samples were slowly cooled in the furnace (Sintered specimens are denoted “HEAP-S”). Some of the sintered specimens were subjected to HIP performed at 1400 °C for 2 h under the pressure of 190 MPa (these specimens are denoted “HEAP-S-HIP”). Another sintered specimens were hot swaged to remove residual porosity (these samples are denoted “HEAP-S-SW”). Specimens were swaged at 950 °C in several consecutive steps with small section reduction with reheating between each step. The same process was also performed using an envelope of beta-titanium alloy in which the specimens were inserted and swaged using the same procedure.

The effect of HIP on specimens directly after CIP was studied as well (the specimens are denoted “HEAP-HIP”). The specimen after HIP was subsequently sintered (1400 °C/14 h/furnace cooled) in order to ensure the chemical homogeneity (the specimen is denoted as “HEAP-HIP-S”).

MA and AT powders were compacted by SPS in an SPS 10-4 device (Santa Rosa, California, CA, USA) using a graphite die (Ø 20 mm) and punches to compact the powder using pressure of 80 MPa. The SPS processing was performed in vacuum. Tungsten foil was used to separate the samples from the graphite tools. The sample was heated with the heating rate of 700 °C/min up to the sintering temperature of 1300 °C at which it was kept for 2 min.

Contrary to CIP followed by HIP, sintering (or swaging), which is suitable for fabrication of large samples, produces relatively small disc shape samples with diameter of 20 mm and thickness of a few mm. Such samples can be strained by high pressure torsion (HPT) [35] in order to refine their structure. Severe plastic deformation applied during HPT processing results in many materials into an extreme grain refinement down to the nanoscale. In the present work HPT straining was performed at room temperature using pressure of 2.5 GPa and 15 revolutions. The samples produced using this route are denoted MA-SPS-HPT and AT-SPS-HPT, respectively. 

The microstructure of specimens was studied by light microscopy (LM), using a Nikon EPIPHOT 3000 (Nikon, Melville, New York, USA) microscope. Scanning electron microscopes (SEM) JEOL 7650F (JEOL, Akishima, Tokyo, Japan) and FEI Quanta 200F (FEI, Hillsboro, OR, USA) were used for microstructural observations as well. The SEM observations were done in back scattered electron mode (BSE) unless otherwise noted. The mean particles size was determined by the analysis of SEM micrographs. NIS Elements software was used for image analysis. Energy dispersive X-ray spectroscopy (EDS) was employed for used analysis of chemical composition. Specimens for LM and SEM observations were prepared by a standard metallographic process (ground up to #4000 with SiC papers and polished with Struers OP-S emulsion with the addition of H_2_O_2_). For etching, 3 mL of HF + 8 mL of HNO_3_ + 100 mL of H_2_O etchants were used. The phase identification was also carried out using the X-ray diffraction analysis (XRD), on a Bruker D8 Discover diffractometer (Bruker, Karlsruhe, Germany) using Cu Kα radiation. XRD investigations were performed in the symmetrical Bragg–Brentano geometry. Quantitative Rietveld refinement analysis was performed by TOPAS V5 software (Bruker, Karlsruhe, Germany).

Vickers hardness was determined using a Zwick/Roell ZHU 250 top hardness tester (Zwick/Roell, Ulm, germany) with the load of 98.1 N (according to the ISO 6507 standard). At least seven values were determined for each measurement. The oxygen content in samples studied was determined by a Bruker Galileo G8 gas fusion analyzer (Bruker, Karlsruhe, Germany). At least three measurements for each specimen were performed. Residual porosity in the samples was determined by the image analysis of SEM micrographs.

Positron lifetime (LT) spectroscopy was employed for characterization of lattice defects in the samples. A ^22^Na radioisotope with activity of 1 MBq deposited on a 2 μm thick mylar foil was used as a positron source. LT measurements were carried out on a digital spectrometer (Hamamatsu Photonics, Hamamatsu, Japan) [36] with time resolution of 145 ps (FWHM of the resolution function). At least 10^7^ positron annihilation events were collected in each LT spectrum. The source contribution to the LT spectra consisted of two components with lifetimes of 368 ps and 1.5 ns and corresponding relative intensities of 11% and 1%, representing contributions of positrons annihilating in the ^22^Na source spot and the covering mylar foil, respectively.

## 3. Results

### 3.1. Initial Powders

The initial powders used for further processing were characterized as their properties may influence the final product.

#### 3.1.1. HEAP Green Compact

Figure 1a shows the microstructure of the initial green compact (HEAP). The green compact obviously consisted of multiple phases. The XRD diffraction pattern for the green compact is plotted in Figure 2. The sample contained the bcc (Nb and Ta powders have both the bcc lattice with similar parameters) and hcp phases (Hf, Ti, and Zr elements). The oxygen content determined in the samples studied is listed in Table 2. The HEAP green compacts exhibited an oxygen concentration of ≈0.55 wt %.

Results of LT measurements are listed in Table 2. The HEAP sample exhibited a two-component LT spectrum. The short component with lifetime τ_1_ represented a contribution of free positrons (not trapped at defects) while the longer component with lifetime τ_2_ could be attributed to positrons trapped at vacancy-like misfit defects at interfaces between precipitates and the matrix and at dislocations introduced into the sample by CIP. Assuming that the component τ_2_ arose exclusively from positrons trapped at dislocations, the mean dislocation density ρ_D_ ≈ 1.48 × 10^14^ m^−2^ in the sample could be calculated from the two-state trapping model [37]. This value should be considered as an upper level of the actual dislocation density in the sample since some fraction of positrons contributing to the component τ_2_ was trapped at misfit defects. This was indicated by the lifetime of trapped positrons value of τ_2_ ≈ 165 ps, which was slightly lower than the value determined for dislocations in the HfNbTaTiZr alloy [38]. 

#### 3.1.2. AT Powder

The microstructure of gas atomized powder (AT) is shown in Figure 1b. The particle sizes of the powder fell into a broad range from 10 to 300 μm. One can see in Figure 1b that the AT particles exhibited a dendritic structure typical for conventionally cast HfNbTaTiZr alloy [39] and consisted of two chemically different phases with bcc structures: one bcc phase, denoted bcc1, was slightly rich in Nb and Ta, and the second bcc phase, denoted bcc2, was rich in Hf, Zr. The XRD pattern for the AT powder is plotted in Figure 3. The two bcc phases had very similar lattice parameters so that their XRD reflections overlapped each other, making the reflections asymmetrical. The lattice parameter *a* = 3.4024(5) Å was determined by Rietveld refinement. The AT powder exhibited a single component LT spectrum with a lifetime of ≈165 ps. It represented a contribution of positrons trapped at misfit defects at the interfaces between bcc1 and bcc2 phases. Since bcc1 and bcc2 phases had slightly different lattice parameters, the discontinuity of lattice planes resulted in the formation of misfit defects at interfaces between these two phases [27]. 

AT powder exhibited low oxygen content (ten times lower than MA) since crucible-free gas atomization was performed in a protective Ar atmosphere and particles of AT powder were rather coarse, i.e., the surface-to-volume ration was relatively low.

#### 3.1.3. MA Powder

The XRD diffraction pattern for MA powder is shown in Figure 3 as well. The sample exhibited very broad XRD reflections at positions corresponding to the bcc phase with lattice parameter *a* = 3.409(1) Å, which was slightly higher than that for the AT powder. Broad XRD reflections testified that MA powder had a nanocrystalline structure. The average size of coherently scattering domains determined by Rietveld refinement was 22.7(6) nm.

The MA powder exhibited rather high oxygen content of ≈1.07 wt %, which was ten times higher than that in the AT powder. Oxygen was incorporated into the MA powder obviously during high energy ball milling due to high surface area of nanocrystalline powder particles. Note that LT investigations of nanocrystalline MA powder were not performed because of its high reactivity with oxygen.

### 3.2. Effect of HIP and Sintering

Figure 4a,b show the microstructure of the HEAP-S specimen. Sintering resulted in equiaxed and relatively fine grains. The chemical homogeneity was good after 16 h sintering as determined by SEM and EDS microstructure analysis. Secondary phases (needle-like precipitates) were observed in the microstructure along with irregular shaped particles of the hcp phase. HIP processing of sintered specimens (HEAP-S-HIP) introduced no significant changes. The microstructure of green compact subjected to HIP treatment (HEAP-HIP) is shown in Figure 4c,d. The chemical homogeneity of HEAP-S was better than that of HEAP-HIP (cf. Figure 4a,c). Several chemical inhomogeneities (i.e., areas with higher Ta, or Nb concentration-probably at the positions of original Ta particles) were observed in the HEAP-HIP sample, as seen in Figure 4c (see EDS maps of typical region in HEAP-HIP specimen in Appendix A). Therefore additional sintering (1400 °C/14 h) was performed resulting in a HEAP-HIP-S specimen with good chemical homogeneity. However, porosity in the HEAP-HIP-S sample remained present in the specimen, and it still consisted of bcc1, bcc2, and hcp phases.

A significant number of pores was observed in the microstructure. The approximate porosity values in the samples studied are listed in Table 2. The as-sintered specimen (HEAP-S) exhibited the highest porosity. The HIP process caused a slight decrease in porosity, but the specimens still remained rather porous, as seen in Figure 4c.

The oxygen content in the initial powder used for sintering (HEAP sample) was relatively high (~0.55 wt %), as seen in Table 2. However, it could be seen that it increased during further processing. The oxygen content increased from 0.55 wt % to 0.65 wt % during sintering. HIP caused even more significant increase in the oxygen content, up to ~0.80 wt %. 

The XRD diffraction patterns of HEAP specimens subjected to HIP and sintering are plotted in Figure 2. From inspection of the Figure 2 it becomes clear that HIP led only to slight broadening of XRD reflections and peak separation (c.f. samples HEAP and HEAP-HIP). This was caused by the fact that the constituting elements were not fully dissolved during the HIP process of green powders (see Figure 4c,d). On the other hand the sintering led to a quite homogeneous microstructure, which resulted in an increase of the concentration of the bcc2 phase at the expense of bcc1. The sintering caused narrowing of XRD profiles, which indicated grain growth occurring during sintering. LT investigations revealed that sintering reduced dislocation density and also the concentration of misfit defects, which was indicated by increases of τ_2_ towards the value of ≈180 ps, typical for dislocations in HfNbTaTiZr alloy [38]. The latter effect was obviously caused by grain growth during sintering. The HIP treatment of sintered sample led to a slight increase of dislocation density, as seen in Table 3, due to dislocations introduced by HIP processing. 

Mechanical properties of samples studied were represented by hardness values, which are listed in Table 2. One can see in the table that hardness of HEAP-S and HEAP-S-HIP specimens were similar (both around 320 HV10). Interestingly the hardness of the HEAP-HIP specimen was around 150 HV10, which was significantly lower value than that of HEAP-S. The hardness of the HEAP-HIP specimen significantly increased during sintering (1400 °C/14 h) to 225 HV10, but did not reach the hardness of sintered specimens. On the other hand the HEAP-S-HIP specimen had a hardness comparable to that of the sintered specimen.

### 3.3. Effect of Swaging

The as-sintered specimen (HEAP-S) was hot swaged to remove residual porosity. However, the swaging process was not completed due to cracking of the specimens, which can be clearly seen in Figure 5. The microstructure of the swaged sample was similar to that of the HEAP-S specimen. The cracking of specimens during hot swaging was at first attributed to a decrease of temperature during contact of the die with the specimen. Due to this reason the specimens were enveloped in a titanium alloy and subsequently swaged in order to ensure sufficiently high temperature of the whole specimen during the process. Unfortunately, the same result was obtained and the specimens were damaged, i.e., cracking occurred as well. It should be pointed out that this specimen contained relatively high amount of oxygen (~0.75 wt %), which may have caused the brittleness of this alloy at the given deformation conditions (i.e., temperature and strain rate). It is also possible that the deformation temperature was too low and the strain rate was too high for the given specimens.

Porosity was not measured in the HEAP-S-SW specimen because the swaging process was interrupted due to specimen cracking, and the distribution of porosity was inhomogeneous. Nevertheless it seems that some regions in the swaged samples had significantly lower porosity and therefore swaging had some potential to eliminate the porosity. Swaging caused a slight increase of the oxygen concentration as well, as seen in Table 2. 

Table 2 shows that hardness increased during hot swaging to more than 400 HV10, indicating strengthening of the material by dislocations introduced by swaging. A significant increase of dislocation density up to ≈3 × 10^14^ m^−2^ in the swaged sample was confirmed by LT spectroscopy, as seen in Table 3. However, it should be accentuated that the specimen was damaged during swaging and therefore this hardness value may not be representative. If the indent was made in the region containing a crack, then it became larger due to a lateral shift of the material caused by opening of the crack. As a consequence the determined hardness value was lower than in the region without the crack. 

### 3.4. Effect of SPS and HPT Processing

The microstructure of the MA-SPS-HPT specimen consisted of an equiaxed very fine (grain size ~4 μm) microstructure with numerous precipitates and some amount of porosity (see Figure 6). It should be accentuated that the MA-SPS specimen was damaged during the HPT process, as many cracks emerged, and the specimen was separated into several smaller pieces. The microstructure of this specimen shown in Figure 6 was obtained from one of these pieces. Figure 6a presents an overview of the microstructure while Figure 6b shows detail of the microstructure with higher magnification. Pores are marked as “1” in the Figure. The particle labelled “2” is the hcp phase rich in Hf and Zr. The bcc1 phase (Nb and Ta rich) distributed mainly on grain boundaries is denoted as “3”. The particles labeled “4” are HfO_2_ oxides (with some minor other elements concentrations). The XRD pattern of the MA-SPS-HPT sample is plotted in Figure 3. The sample contained a bcc phase with lattice parameter *a* = 3.404(5) Å and a small concentration of HfO_2_ oxide.

The MA-SPS-HPT sample had the highest oxygen concentration among the samples studied, as seen in Table 2, which indicates that a significant concentration of oxygen was introduced into the alloy during the MA processing.

The AT-SPS-HPT sample had an equiaxed microstructure with a pure bcc phase and the lattice parameter *a* = 3.406(4) Å. No residual porosity and no precipitates of secondary phases were observed, as seen in Figure 7. The mean grain size of sample AT-SPS (i.e., before HPT straining) was ≈50 μm, as seen in Figure 7. After HPT deformation (sample AT-SPS-HPT), the mean grain size decreased below 0.5 μm. The microstructure was highly deformed. The AT-SPS-HPT samples contained the lowest concentration of oxygen among the all samples studied (~0.12 wt %). Hence, the SPS treatment resulted only in a slight increase of oxygen content, if any.

One can see in Table 3 that both MA-SPS-HPT and AT-SPS-HPT samples exhibited a single component LT spectrum with lifetime τ_2_ ≈ 180 ps. This testifies that virtually all positrons were annihilated in a trapped state in dislocations (so called saturated positron trapping). Hence, one can conclude that HPT processing resulted in formation of a high density of dislocations exceeding 5 × 10^14^ m^−2^.

## 4. Discussion

### 4.1. Residual Porosity

Results obtained for HEAP-S and HEAP-S-HIP samples (see Table 2) indicate that HIP processing had some potential to decrease residual porosity. However, this decrease was quite small and considerable porosity remained in the specimen. The fact that even HIP processing at 1400 °C/200 MPa/2 h did not remove the porosity completely can be ascribed to a high volume fraction of pores and to filling of the pores by Ar gas used as a pressing medium in HIP processing. The MA powder compacted by SPS contained residual porosity as well, and small pores remained in the sample even after HPT processing, as seen in Figure 6. It should be pointed out that the MA-SPS-HPT specimen was broken into several pieces during the HPT process and therefore the results of such specimen are disputable. In such cases, the HPT after cracking could not introduce full deformation (strain) to the specimen. Contrary to this, SPS processing at 1300 °C of AT powder resulted in porosity-free samples (see Figure 7). 

It has to be mentioned that no microscopic porosity (vacancy clusters or voids) was found by LT investigations in the HEA samples studied in this work, which means that residual porosity in the present samples was mainly macroscopic porosity (large pores among grains).

### 4.2. Oxygen Content

The oxygen concentration in specimens fabricated by powder metallurgy techniques was determined not only by oxygen content in the initial powders, but also by the processing route. The highest oxygen concentration was found in the MA-SPS specimen. This was probably caused by a significant oxygen contamination of powders with large active surface area. This was possibly due to a long milling time and residual oxygen present in the milling atmosphere (Ar), and high reactivity of some elements with the oxygen (e.g., Ti, Hf) [40]. In the HEAP samples the oxygen content increased more significantly during HIP (Ar atmosphere) than during sintering (vacuum <1.10^−3^ Pa). It also seems that the oxygen content was decisive for the plasticity of the alloy as the only specimen with enough ductility to undergo the cold plastic deformation was the AT-SPS specimen, which had the lowest oxygen content. Other specimens were damaged by crack formation during deformation at room temperature or even at higher temperature (swaging at 950 °C). It should be pointed out that the possibility of hot deformation of the alloy with increased oxygen content cannot be excluded under suitable conditions (i.e., low strain rate, or higher processing temperature) [41], but this possibility needs more detailed study. 

The high oxygen content in the MA-SPS alloy well corresponds to the presence of numerous oxides (mainly HfO_2_) in the microstructure detected by SEM and by XRD. They emerged probably during mechanical alloying and are also one of the reasons of the brittleness of the alloy [42]. The thermal stability of these oxides is very high and therefore they were not dissolved during the SPS process [43]. The second reason is probably the oxygen dissolved in the matrix [44]. The AT-SPS is the only sample with single phase bcc solid solution with no other phases. Single phase random solid solution is caused by relatively high cooling rate after SPS processing at high temperature of 1300 °C, where all secondary phases are supposed to be dissolved [2,22,26,45,46,47,48]. It was reported by Senkov et al. [22] that cooling rate higher than 15 °C/min is sufficient for retaining single phase random solid solution at this alloy. The cooling rate after SPS is very high as the sample was cooled from sintering temperature (1300 °C) to room temperature in 3–4 min. Therefore the achieved cooling rate around 300 °C/min is much higher than 15 °C/min. On the other hand the specimens after sintering or HIP were slowly cooled with the furnace. The estimated cooling rate (HIP and sintering had similar cooling rate) was lower than 12 °C/min under 800 °C (lowering with decreasing temperature). This cooling rate was not high enough to suppress formation of the bcc2 and hcp phases. 

The grain size of the MA-SPS specimen was lower than for the others (except that after HPT). This was caused by the intensive plastic deformation during high energy ball milling, which resulted in a high number of nucleation sites for new grains formed by recrystallization during SPS processing.

### 4.3. Hardness

Very high hardness of the MA-SPS-HPT specimen (584 HV10) can be ascribed to the high oxygen content in solid solution and also the presence of oxide particles that caused additional strengthening. As seen in Figure 8 hardness was correlated with oxygen content in the sample. Hardness remained approximately unchanged with increasing oxygen content up to the oxygen concentration of ≈0.8 wt %. Above this value hardness strongly increased with oxygen concentration. This indicates that there was a certain threshold of oxygen concentration in the sample above which it had a significant hardening effect. It is likely that the remarkable hardening effect of oxygen above 0.8 wt % was connected with the formation of HfO_2_ particles with a monoclinic structure, which were detected in MA-SPS and MA-SPS-HPT samples by SEM and X-ray diffraction. Additional strengthening (not connected with oxygen) was caused by dislocations introduced by severe (cold) plastic deformation. In Figure 8 one can see that HPT straining increased hardness of both MA-SPS-HPT and AT-SPS-HPT samples. 

It has to be mentioned that HEAP-HIP and HEAP-HIP-S samples were characterized by surprisingly low hardness. Hardness measurement of these samples could be influenced by porosity. Moreover, the low hardness of the HEAP-HIP sample could be possibly attributed to different phase composition. One can see in Figure 4c that Ta and Nb rich particles were not fully dissolved during HIP and therefore this specimen was somewhat between a blend of powders and a high entropy alloy. This was proved also by the XRD pattern in Figure 2, which showed the highest difference between lattice parameters of the bcc1 and bcc2 phases. It was reported that the high entropy effect may result into higher hardness since various radii of atoms occupying randomly lattice sites cause random fluctuations of potential energy landscape, which hinders motion of dislocations [2]. This hardening effect was probably absent in the HEAP-HIP sample, as random solid solution was not completely formed. This picture is supported by the fact that both bcc1 (Nb, Ta rich) and bcc2 (Hf and Zr rich) phases have lower hardness than HfNbTaTiZr random solid solution [38]. Subsequent sintering improved the chemical homogeneity of the HEAP-HIP sample and increased hardness as well. However, the hardness of HEAP-HIP-S still did not reach the values of HEAP-S and HEAP-S-HIP. The cause of this effect remains unclear to the authors.

## 5. Conclusions

HfNbTaTiZr high entropy alloy was produced by various methods of powder metallurgy including mechanical alloying, sintering, hot isostatic pressing, spark plasma sintering, and high pressure torsion. The microstructure and mechanical properties of HfNbTaTiZr prepared by various methods were examined. It was found that the main problem in fabrication of HfNbTaTiZr alloy by powder metallurgy is oxygen contamination. Samples prepared by cold isostatic pressing followed by sintering and/or hot isostatic pressing contained oxygen in the concentration range of 0.55 wt %–0.85 wt %. Oxygen contamination of samples prepared by mechanical alloying was even higher, and the oxygen concentration in these samples exceeded 1 wt %. Oxygen incorporated into the HfNbTaTiZr samples caused deterioration of mechanical properties (embrittlement) and prevented formation of a single phase random solid solution. 

Investigations performed in the present work showed that the most promising powder metallurgy method of production of HfNbTaTiZr alloy was spark plasma sintering of gas atomized powder. HfNbTaTiZr alloy prepared using this route was found to be a single phase random solid solution, and oxygen content in the sample was ≈0.12 wt % only. Moreover, it was shown that high pressure torsion is a suitable method for grain refinement of high entropy alloys prepared by powder metallurgy.

## Figures and Tables

**Figure 1 materials-12-04022-f001:**
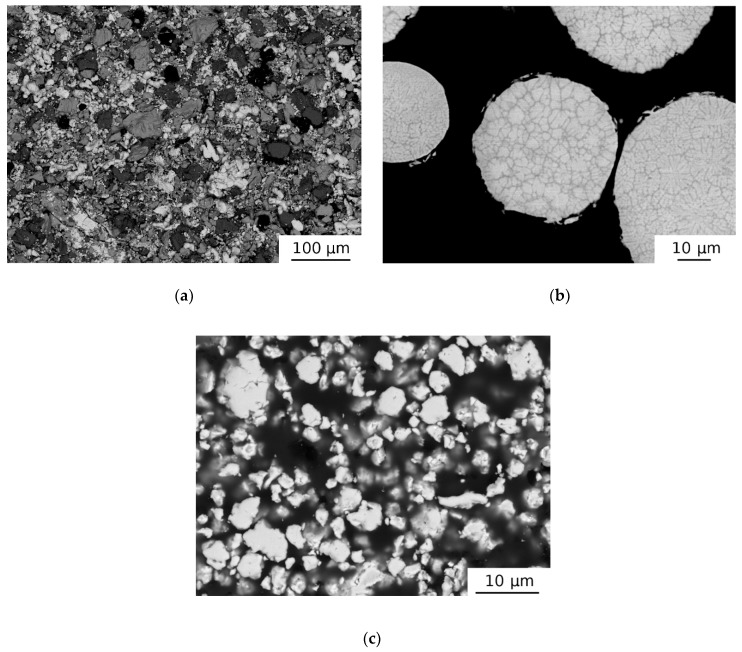
SEM micrographs showing microstructure of (**a**) HEAP green compact, (**b**) AT initial powder, and (**c**) MA initial powder.

**Figure 2 materials-12-04022-f002:**
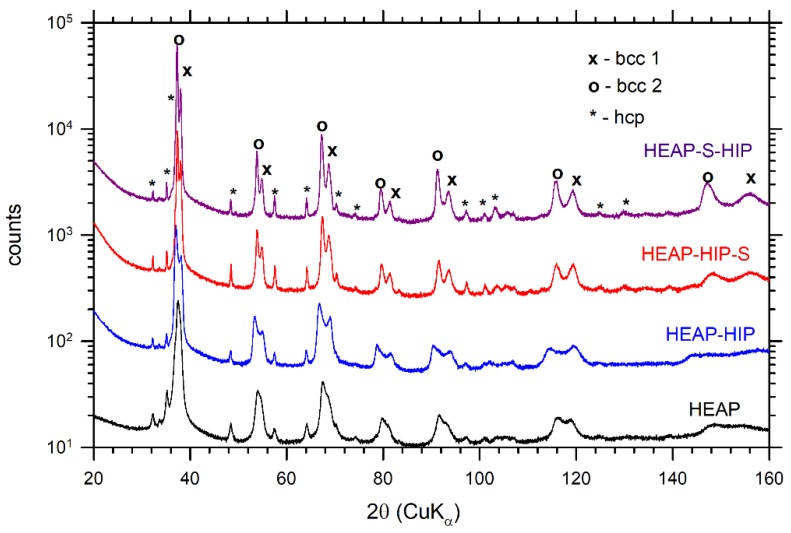
XRD patterns for HEAP, HEAP-HIP, HEAP-HIP-S, and HEAP-S-HIP specimens. The samples contain two bcc phases (bcc1, bcc2) and a hcp phase. The XRD patterns were shifted vertically for better visibility.

**Figure 3 materials-12-04022-f003:**
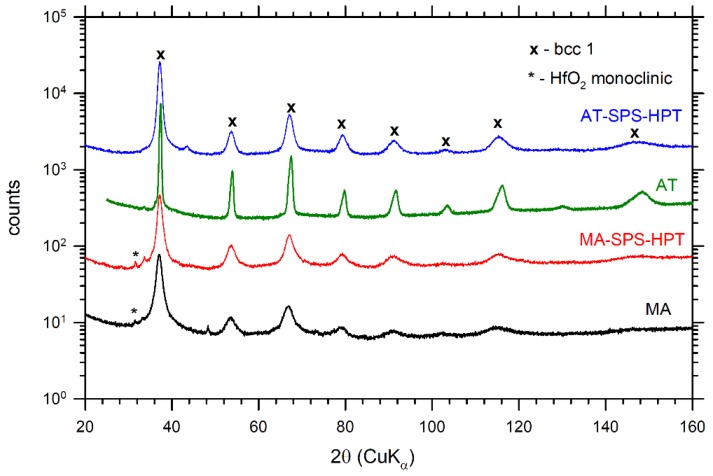
XRD patterns for samples AT, MA, AT-SPS, MA-SPS, AT-SPS-HPT, and MA-SPS-HPT. The samples contain a bcc phase and a small fraction of a monoclinic HfO_2_ phase.

**Figure 4 materials-12-04022-f004:**
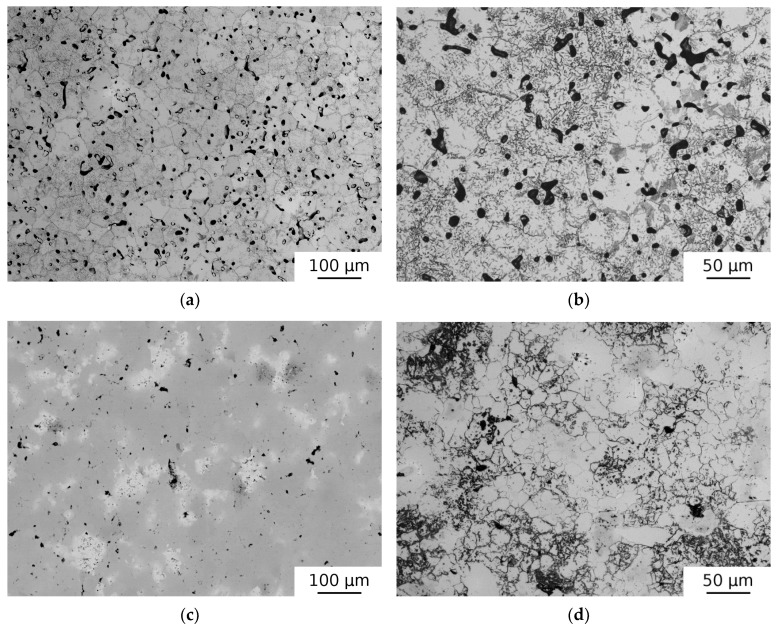
Micrographs of (**a**) HEAP-S, (**b**) HEAP-S—detail of etched specimen, (**c**) HEAP-HIP (SEM-BSE), and (**d**) detail of HEAP-HIP (LM-etched).

**Figure 5 materials-12-04022-f005:**
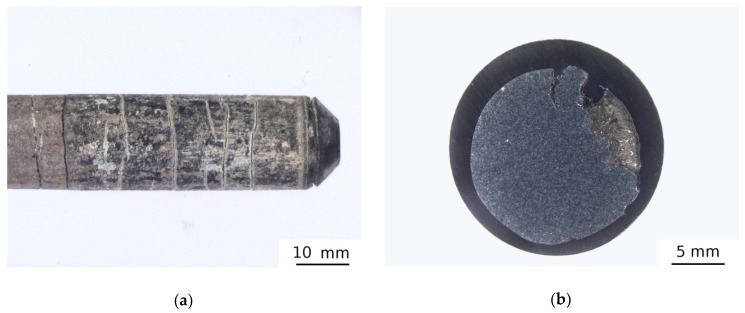
The microstructure of the hot swaged (HEAP-S-SW) specimen: (**a**) overview with radial cracks; (**b**) after forging in Ti-envelope.

**Figure 6 materials-12-04022-f006:**
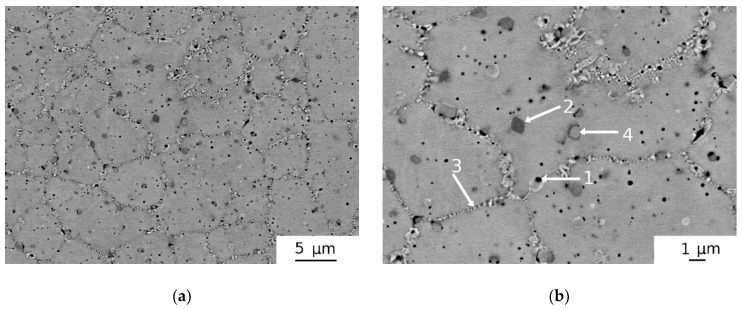
SEM micrograph of MA-SPS-HPT specimen: (**a**) overview of microstructure; (**b**) detail of microstructure with the pores (marked “1”) and three kinds of precipitates (marked “2, 3, 4”).

**Figure 7 materials-12-04022-f007:**
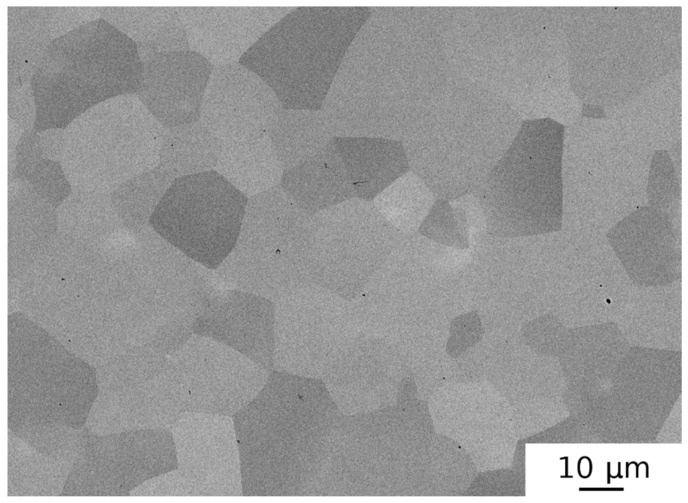
SEM micrograph of AT-SPS specimen.

**Figure 8 materials-12-04022-f008:**
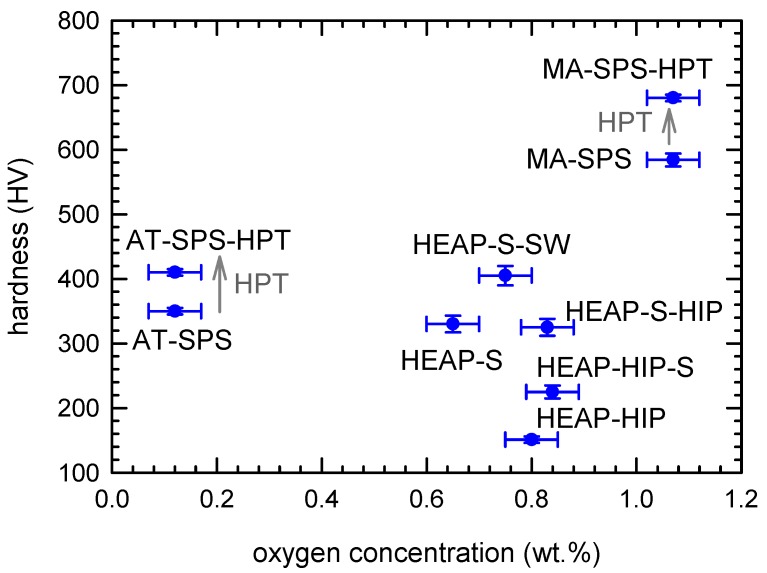
Correlation between hardness of samples studied and oxygen content. Grey arrows denote work hardening effect of HPT straining.

**Table 1 materials-12-04022-t001:** Specimens and their processing routes.

Specimen	Process Route
HEAP	CIP
HEAP-S	CIP + sintering
HEAP-S-HIP	CIP + sintering + HIP
HEAP-HIP	CIP + HIP
HEAP-HIP-S	CIP + HIP + sintering
HEAP-S-SW	CIP + sintering + hot swaging
MA-SPS	mechanical alloying + SPS
AT-SPS	atomized powder + SPS
MA-SPS-HPT	mechanical alloying + SPS + HPT
AT-SPS-HPT	atomized powder + SPS + HPT

**Table 2 materials-12-04022-t002:** Properties (porosity, grain size, hardness, and oxygen content) of studied specimens.

Specimen	Porosity (%)	Grain Size (μm)	Hardness (HV10)	Oxygen (wt %)
HEAP	N/A	N/A	N/A	0.55 ± 0.05
HEAP-S	6.5 ± 1	~35	330 ± 13	0.65 ± 0.07
HEAP-S-HIP	4.5 ± 0.5	~45	325 ± 13	0.83 ± 0.02
HEAP-HIP	5 ± 0.5	~20	151 ± 7	0.80 ± 0.05
HEAP-HIP-S	5 ± 1	~25	225 ± 10	0.84 ± 0.06
HEAP-S-SW	N/A^1^	N/A	405 ± 15^2^	0.75 ± 0.03
MA-SPS	0.9 ± 0.3	~10	584 ± 10	1.07 ± 0.10
AT-SPS	0	~50	350 ± 5	0.12 ± 0.02
MA-SPS-HPT	N/A^1^	~4	680 ± 5^2^	1.07 ± 0.03
AT-SPS-HPT	0	~0.5	410 ± 5	0.12 ± 0.02

^1^ Porosity was not measured due to damaged specimens. ^2^ Hardness of HEAP-S-SW and MA-SPS-HPT is disputable due to defects in the microstructure (e.g., cracks).

**Table 3 materials-12-04022-t003:** Results of LT spectroscopy: lifetimes τ_i_ and relative intensities I_i_ of the components resolved in LT spectra. The mean density of dislocations ρ_D_ calculated using the two-state simple trapping model is shown in the Table as well.

Specimen	τ_1_ (ps)	I_1_ (%)	τ_2_ (ps)	I_2_ (%)	ρ_D_ (10^14^ m^−2^)
HEAP	69(2)	12(1)	165(2)	88(1)	1.48(8)
HEAP-S	85(2)	22(2)	175(4)	78(2)	0.94(9)
HEAP-S-HIP	81(4)	20(2)	178(3)	80(2)	1.1(1)
HEAP-HIP	81(3)	17(3)	168(3)	83(3)	1.1(1)
HEAP-HIP-S	86(4)	23(2)	176(4)	77(2)	0.9(1)
HEAP-S-SW	45(4)	9(1)	181(2)	91(2)	3.0(1)
AT	-	-	165(2)	100	<0.1
MA-SPS	80(6)	8(1)	150(2)	92(1)	1.1(1)
AT-SPS	148(1)	100	-	-	<0.01
MA-SPS-HPT	-	-	180(1)	100	>5
AT-SPS-HPT	-	-	183(1)	100	>5

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
