# Peer review of "The Effect of Processing Route on Properties of HfNbTaTiZr High Entropy Alloy"

_materials, 2019, doi:10.3390/ma12234022_

Round 1
Reviewer 1 Report
The article presents comprehensive investigations and interesting results. Therefore, the article is suitable for the publication in MDPI Materials. However, the article requires intensive revision. The spelling and formatting has to be improved. Furthermore, partial restructuring is required. Especially 3.1.1 - 3.1.2 need revision and a stronger focus on the actual heading.
Detailed comments can be found in the attached file.

Author Response
Reviewer 1
Comment:
The article presents comprehensive investigations and interesting results. Therefore, the article is suitable for the publication in MDPI Materials. However, the article requires intensive revision. The spelling and formatting has to be improved. Furthermore, partial restructuring is required. Especially 3.1.1 - 3.1.2 need revision and a stronger focus on the actual heading.
Response:
Spelling and formatting of the paper was improved. In particular the sections 3.1.1 and 3.1.2 were extensively rewritten in order to improve clarity and focus on the actual heading.
Comment:
Material and Methods Full experimental details have to be stated
Response:
Full experimental details for each method of HEA preparation were added to the revised manuscript.
Comment:
Line 84: Mixing of powder for 10 hours (wrong unit??)
Response:
Mixing of elemental powders in a Turbula 2F device was performed indeed for 10 hour.
The time of mixing was based on our previous results with similar powders (-325 mesh) used for beta-titanium alloys). This time was proved to ensure excellent homogeneity of powders (some powder agglomerates were observed after shorter mixing periods in titanium alloys.
Comment:
Line 103: which powder particle size / fraction has been used for SPS
Response:
The mean particle size of MA powder used for SPS was 4 mm. The AT powder has a broad size distribution of particles covering the range from 10 to 300 mm. This information was added to the revised manuscript (lines 89-93).
Comment:
Line 107: why two different post treatments (swaging; HPT) have been used for different samples? This impedes direct comparability
Response:
Swaging treatment can be used for massive samples (in the present case rods with diameter of 15 mm, but larger specimens can be relatively easily processed) and can be therefore relatively easily used in industrial applications. On the other hand, HPT allows achieving extremely high strength but can be used for a small disk shape samples only. In the present case HPT treated samples have disc shape with diameter of 20 mm and thickness of 2 mm. Since CIP followed by sintering and/or HIP enables production of massive samples swaging was considered as a suitable post-treatment of these massive samples. On the other hand, by SPS one can produce small disc shape samples which are suitable for HPT processing. This explanation was added to the manuscript (lines 112-117).
Comment:
Line 119: it is unclear which samples have been etched
Response:
Samples studied by LM were etched. If the specimen was etched it is stated in figure caption in revised manuscript. In some cases polishing with Strued OP-S emulsion with the H2O2 addition may also cause light etching effect.
Comment:
How was the oxygen content determined?
Response:
The oxygen content was determined by a Bruker Galileo G8 inert gas fusion analyzer. This information is mentioned in the manuscript on the lines 135.
Comment:
How was the particle size measured (image analysis)
Response:
Particle size was determined by image analysis of SEM and light microscopy images. This information was added to the revised manuscript (lines 123-124).
Comment:
How was the porosity measured
Response:
Residual porosity in the samples was determined by image analysis of SEM micrographs. Measuring of porosity was performed by image analysis rather than via Archimedes test (measuring density) as due to certain amount of open porosity the Archimedes test did not give good results. This information was added to the revised manuscript (lines 136-137).
Comment:
Results Restructuring Line 161: how was the particle size measured
Response:
The particle size was measured by analysis of SEM micrographs. This information was added to the revised manuscript (lines 123-124).
Comment:
1.3 why are there no images of the MA powder
Response:
SEM of image of MA powder was added to the manuscript as Fig. 1c.
Comment:
All oxygen contents are stated with two digits, have they been measured by EDX?
Response:
Oxygen content was determined by inert gas fusion analyzer Bruker Galileo G8. The value is an average from at least 3 values for each specimen. This is mentioned in the section 2, line 135.
Comment:
Line 192: ??? phase
Response:
The misprint on line 208 was corrected. ??? phase -> monoclinic HfO2 phase.
Comment:
Line 195: “the chemical homogeneity is good after 16 hours…”, how was this evaluated?
Response:
This was evaluated by SEM microstructure observations combined with EDS chemical analysis. This information was added to the revised manuscript (lines 211-212).
Comment:
Line 261: “damaged microstructure can influence the measurement” ??
Response:
This statement has been clarified in the revised manuscript.
If the indent is made in the region containing crack then it becomes larger due to shift of the material caused by opening of the crack and the measured hardness value is therefore lower. As a consequence the measured hardness value depends whether the indent was made in the region containing crack or not. This explanation was added to the revised manuscript (lines 274-277).
Comment:
Discussion Line 336-338: Misleading statement; cooling rate to slow?; has the cooling rate been determined for sintering and HIP?
Response:
The misleading statement about cooling rate has been corrected:
“On the other hand the specimens after sintering or HIP were slowly cooled with the furnace. The estimated cooling rate was lower than 12oC/min which is not high enough to suppress formation of the bcc2 and hcp phases.” In this case the value 12°C/min is at around 800°C. At lower temperatures it decreases. In the given alloy the temperatures between 800°C and 600°C are supposed to be the most important for bcc2 phase formation.
Comment:
Figure 8: insufficient discussion; what does the line represent? (better omit)
Response:
Discussion related to Figure 8 has been improved and extended in the revised manuscript (lines 356-367). Solid line in Figure 8 was intended as guide of eyes only. It has been omitted in the revised manuscript.
Reviewer 2 Report
The manuscript analyzes the effect of processing route on the microstructure, density and homogeneity of HfNbTaTiZr alloy. The manuscript presents results which could be of interest to the materials research community, but several improvements need to be made to the manuscript before it can be considered for publication.
The manuscript is not well written. It consists of typographical and grammatical errors. Also, the flow of the manuscript is difficult to follow. For example, in the “Effect of HIP and sintering” section, Figure 4a is referred in the first paragraph and some microstructural aspects have been described. Again, in the sixth paragraph of the same section, figure 4 has been referred and same aspects have been dealt with. Such abrupt jumps disrupt the flow of the manuscript. It is claimed that the HIP treatment could not reduce porosity in the HEAP-S-HIP samples because of the presence of open pores in HEAP-S samples. But according to the SEM micrograph shown in Fig. 4a and 4c, the pores do not appear to be open pores. These pores should have been eliminated by HIP. This requires better clarification. Caption for Figure 3 needs to be fixed. Porosity for MA-SPS-HPT is shown to be 0% in Table 3, but clearly, Figure 6 shows porosity. Why is the data misrepresented? It is mentioned that HIP treatment of HEAP-S samples results in an increase in dislocation density. This should lead to an increase in hardness. Why does the hardness decrease after HIP treatment? Simply stating the results might not be sufficient. These are significant results and therefore need explanation. It is mentioned that chemical inhomogeneities were observed in HEAP-HIP samples and these were identified as areas with higher Ta, or Nb concentration. How was it determined that these are areas with higher Ta or Nb?Author Response
Reviewer 2
The manuscript analyzes the effect of processing route on the microstructure, density and homogeneity of HfNbTaTiZr alloy. The manuscript presents results which could be of interest to the materials research community, but several improvements need to be made to the manuscript before it can be considered for publication.
Comment:
The manuscript is not well written. It consists of typographical and grammatical errors. Also, the flow of the manuscript is difficult to follow. For example, in the “Effect of HIP and sintering” section, Figure 4a is referred in the first paragraph and some microstructural aspects have been described. Again, in the sixth paragraph of the same section, figure 4 has been referred and same aspects have been dealt with. Such abrupt jumps disrupt the flow of the manuscript.
Response:
English was improved in the whole paper. Typographical errors have been corrected as well. Clarity of the manuscript was improved in order to make it easier to follow by the reader.
Comment:
It is claimed that the HIP treatment could not reduce porosity in the HEAP-S-HIP samples because of the presence of open pores in HEAP-S samples. But according to the SEM micrograph shown in Fig. 4a and 4c, the pores do not appear to be open pores. These pores should have been eliminated by HIP. This requires better clarification.
Response:
By comparing Fig. 4a and 4b one can conclude that pores are still present in the sample after HIP processing although their density is lower compared to the sample HEAP-S. This can be seen also in Table 1 where the porosity of HEAP-S sample is (6.5 ± 1)% while the porosity of HEAP-S-HIP samples is (4.5 ± 0.5)%. Hence, although HIP processing decreased the concentration of pores it was not able to remove porosity completely. It might be also due to filling of pores by Ar gas used as a pressing medium in HIP. This is now mentioned in the revised manuscript (line 315-316).
The referee is right that these pores are not open pores. The adjective “open” was omitted in the text.
Comment:
Caption for Figure 3 needs to be fixed.
Response:
Caption of Fig. 3 was fixed: small fraction of ??? phase -> small fraction of monoclinic HfO2 phase.
Comment:
Porosity for MA-SPS-HPT is shown to be 0% in Table 3, but clearly, Figure 6 shows porosity. Why is the data misrepresented?
Response:
By a mistake there was 0% porosity value for MA-SPS-HPT sample in Table 3. This mistake was corrected in the revised manuscript. We decided not evaluate the porosity of such specimen as it could be misleading due to the fact that the specimen was damaged (cracked) during the HPT process as is stated in the text. From the given figure it could be deduced that the porosity is similar to that of the MA-SPS specimen. We apologize for leaving the 0 value in the final version of manuscript instead of N/A with the remark.
Comment:
It is mentioned that HIP treatment of HEAP-S samples results in an increase in dislocation density. This should lead to an increase in hardness. Why does the hardness decrease after HIP treatment? Simply stating the results might not be sufficient. These are significant results and therefore need explanation.
Response:
We completely agree with the referee that relatively low hardness of HEAP-HIP and HEAP-HIP-S samples is surprising. HIP processing slightly increased the dislocation density from 0.94 (HEAP-S sample) to 1.1 x 1014 m-2 (HEAP-S-HIP sample), see Table 3. Dislocations cause work strengthening which leads to an increase of hardness. However, in the samples HEAP-HIP and HEAP-HIP-S the hardness is influenced mainly by incomplete dissolution of constituting elements. HIP processing of green compact did not result in random solid solution. As shown in Fig. 4b HEAP-HIP sample contains Ta and Nb rich particles which were not fully dissolved during HIP. Hence, the HEAP-HIP specimen is somewhere between a blend of powders and a high entropy alloy. As a consequence the hardening effect caused by misfit due to different atomic sizes is absent or at least less pronounced in the HEAP-HIP sample. This is likely the reason for lower hardness of the HEAP-HIP sample compared to other samples. This explanation was added to the revised manuscript (lines 372-386).
Comment:
It is mentioned that chemical inhomogeneities were observed in HEAP-HIP samples and these were identified as areas with higher Ta, or Nb concentration. How was it determined that these are areas with higher Ta or Nb?
Response:
The regions with enhanced Ta and Nb concentration in the sample HEAP-HIP (white areas in Fig. 4b) were identified by EDS analysis. It revealed enhanced intensity of Ta and Nb characteristic X-ray peaks in spectra from the white areas in Fig. 4b. Hence, one can conclude that HEAP-HIP sample contains regions enhanced in Ta and Nb.
Round 2
Reviewer 1 Report
The article has been intensively revised according to the previous review. A detailed response to the comments is provided. Therefore the article is suitable for publication.
Reviewer 2 Report
The manuscript has been revised and can now be accepted for publication.